# Maternal and neonatal outcomes in patients with hepatitis C and intrahepatic cholestasis of pregnancy: The sum of the parts

Emily C. Goins[1], Lauren E. Wein[2], Virginia Y. Watkins[2], Alexa I. K. Campbell[1], R. Phillips Heine[3], Brenna L. Hughes[2], Sarah K. Dotters-Katz[2], Jerome Jeffrey Federspiel [2,4] *

1 Duke University School of Medicine, Durham, North Carolina, United States of America, 2 Department of Obstetrics and Gynecology, Division of Maternal Fetal Medicine, Duke University School of Medicine, Durham, North Carolina, United States of America, 3 Department of Obstetrics and Gynecology, Division of Maternal Fetal Medicine, Wake Forest University School of Medicine, Winston-Salem, North Carolina, United States of America, 4 Department of Gynecology and Obstetrics, The Johns Hopkins University School of Medicine, Baltimore, MD, United States of America

* jerome.federspiel@duke.edu

**Data Availability Statement:** Data cannot be shared publicly because of restrictions from the Healthcare Cost and Utilization Project. Data are available from the Healthcare Cost and Utilization

## Abstract

### Objective

Hepatitis C virus and intrahepatic cholestasis of pregnancy (ICP) are well-known independent risk factors for adverse outcomes in pregnancy. In addition, it is well-established that there is an association between Hepatitis C and ICP. This study's objective was to describe the impact of having both Hepatitis C and ICP on maternal and obstetric outcomes compared to patients having either Hepatitis C or ICP.

### Methods

We conducted a retrospective cohort study of the Nationwide Readmissions Database, an all-payor sample of discharges from approximately 60% of US hospitalizations. Deliveries at 24–42+ weeks between 10/2015 and 12/2020 were included. Diagnosis of Hepatitis C and ICP, and outcomes related to severe maternal morbidity were identified using International Classification of Disease-10 codes. Patients were categorized based on Hepatitis C and ICP status. Weighted logistic and negative binomial regression analyses were used to evaluate the association between Hepatitis C and ICP status and outcomes, adjusting for patient and hospital characteristics. The primary outcome was any severe maternal morbidity; secondary outcomes included acute respiratory distress syndrome, acute kidney injury, sepsis, gestational diabetes, cesarean delivery, preterm birth, and hospital length of stay. We modeled interaction terms between ICP and Hepatitis C to assess whether there was a greater or lesser effect from having both conditions on outcomes than we would expect from additive combination of the individual components (*i.e.*, synergy or antagonism).

### Results

A total of 10,040,850 deliveries between 24–42+ weeks were identified. Of these, 45,368 had Hepatitis C only; 84,582 had ICP only; and 1,967 had both Hepatitis C and ICP. Patients

Project for researchers who meet the criteria for access to the data.

**Funding:** The work contained in this manuscript were made possible by the following grants from the National Institutes of Health (TL1-TR002555 [JJF] and K12-HD103083). Data acquisition was also supported by funding from the Foundation for Women and Girls with Blood Disorders to JJF. The funders had no role in study design, data collection and analysis, decision to publish, or preparation of the manuscript.

**Competing interests:** The authors report no conflict of interest.

with both Hepatitis C and ICP had 1.5-fold higher odds of developing severe maternal morbidity compared to having neither. There was an also an increased odds of severe maternal morbidity in patients with both Hepatitis C and ICP compared to patients with only Hepatitis C or ICP. Having both was also associated with higher odds of preterm birth and length of stay compared to having only Hepatitis C, only ICP, or neither (preterm birth: aOR 5.09, 95% CI 4.87–5.33 vs. neither; length of stay: 46% mean increase, 95% CI 35–58% vs. neither). Associations were additive—no significant interactions between hepatitis C and cholestasis were found on rates of severe maternal morbidity, acute respiratory distress syndrome, acute kidney injury, sepsis, cesarean section, or preterm birth (all p>0.05), and was minimal for gestational diabetes and length of stay.

## Conclusion

Hepatitis C and ICP are independent, additive risk factors for adverse maternal and obstetric outcomes. Despite physiologic plausibility, no evidence of a synergistic effect of these two diagnoses on outcomes was noted. These data may be useful in counseling patients regarding their increased risk of adverse outcomes when ICP presents in association with Hepatitis C versus ICP alone.

## Introduction

Hepatitis C virus (HCV) is the most common chronic blood-borne disease in the United States, complicating 421.4 per 100,000 births in 2016 and 463.7 per 100,000 births in 2021 [1,2]. It is associated with increased risk of adverse maternal and neonatal outcomes, including gestational diabetes (GDM), preterm delivery, and small birth weight for gestational age [3]. Similarly, the rate of intrahepatic cholestasis of pregnancy (ICP) among pregnant patients is estimated at 0.3–0.5% [4]. ICP is associated with several similar obstetric complications, including increased risk of pre-eclampsia; in the fetus and neonate, ICP is also associated with stillbirth, preterm delivery, intensive care unit admission, and meconium-stained amniotic fluid [5,6].

A cross-sectional study of pregnant individuals at 4 sites in China demonstrated an association between HCV and ICP, with evidence that HCV infection could increase the risk of developing ICP [7]. Other studies have shown that patients with ICP are at increased risk of more advanced HCV infection and higher HCV viral load [7–9]. The mechanism behind this association is unclear, with some research suggesting HCV may directly increase bile acids in birthing people, and some suggesting a more indirect role through alteration of bile acid metabolism [8,10]. While it is known that each condition is associated with an increased risk of adverse maternal and obstetric outcomes, there are currently no studies investigating the effects of HCV and ICP co-occurrence on outcomes. Our objective was to compare maternal outcomes between birthing individuals with HCV alone, ICP alone, and HCV and ICP together. Given that both diseases target the liver, it may be that these two diseases interact to worsen severe maternal morbidity, beyond the simple additive combination of the two (*i.e.*, that they demonstrate synergy).

## Materials and methods

### Study population

This retrospective cohort study was conducted using the Nationwide Readmissions Database (NRD), from the United States Agency for Healthcare Research and Quality's Healthcare Cost

and Utilization Project. The NRD is an all-payor administrative database designed to represent short-stay inpatient admissions in the United States. A total of 28 states participated in the NRD in 2020, including 59.7% of the population and 58.7% of hospitalizations. Weighting and stratification variables are provided in the NRD dataset to allow estimating national rates of hospitalizations using the NRD sample. The data include patient demographics, hospital characteristics, International Classification of Diseases Diagnosis and Procedure codes, discharge disposition, length of stay, and inpatient charges. While it is possible to follow readmissions for a patient, we limited analyses to the delivery hospitalization.

Deliveries at or beyond 24 weeks gestation with hospital discharges between 10/2015 and 12/2020 were included. The gestational age of 24 weeks was chosen because it is traditionally considered the limit of viability. Pregnant patients were identified based on procedural coding with the International Classification of Diseases 10[th] revision, Clinical Modification (ICD-10-CM) (S1 Table). The starting date was selected given this was when ICD-10 was adopted for clinical use in the United States. Weeks of gestation was determined using the Z3A.xx series of codes. Diagnosis of HCV was identified for each patient using the ICD-10-CM codes for carrier of viral hepatitis C (Z22.52), chronic viral hepatitis C (B18.2), unspecified viral hepatitis C without hepatic coma (B19.20), or unspecified viral hepatitis C with coma (B19.21). Diagnosis of ICP was determined by whether patients had ICD-10 codes for both liver and biliary tract disorder in pregnancy (O26.6*) and obstruction of bile duct (K83.1). We consulted with an experienced inpatient obstetrics coder at our institution in developing these criteria.

## Outcomes

The primary outcome for this analysis was occurrence of severe maternal morbidity (SMM) during the delivery hospitalization, based on criteria and the ICD-10 codes published by the Centers for Disease Control and Prevention (CDC) [11]. Our secondary outcomes included acute respiratory distress, acute kidney injury, sepsis, GDM, cesarean delivery, preterm birth defined as birth at less than 37 weeks gestational age, and hospital length of stay.

## Statistical analysis

The NRD dataset includes weighting and stratification variables to allow for estimating national rates of hospitalizations. For comparing demographics, comorbidities, hospital characteristics, and SMM in each patient cohort, weighted linear regressions were used for continuous variables and weighted chi-square testing was used for binary and categorical variables as appropriate. Weighted logistic (all outcomes except length of stay) and negative binomial (length of stay) regression analyses were used to evaluate the association between HCV and ICP status and outcomes. Models were adjusted for age, primary payer, median household income by ZIP code, hospital type and size, year, discharge quarter, and clinical co-morbidities as identified by the expanded obstetric comorbidity index [12]. Missing values were only present for the ZIP code income (0.4%) and primary payer (0.06%) variables; given the low prevalence of missing values and complications of a weighted dataset, modal value imputation was performed. We assessed for synergistic or antagonistic effects by including an interaction term between ICP and HCV, which was retained in the final statistical models if it was statistically significant for the given outcome. Statistical analyses were performed in SAS Version 9.4 (SAS Institute, Cary, NC) and Stata Statistical Software version 16.1 (StataCorp, College Station, Texas). A two-sided alpha level of 0.05 was pre-specified as statistically significant. Because the NRD dataset is a Limited Data Set as defined by the Health Insurance Portability and Accountability Act, this study was ruled exempt from review by the Duke University Health Institutional Review Board (Pro00106911). As a Limited Data Set, the NRD dataset does not provide

information sufficient to contact and obtain informed consent from participants (and in fact this would be prohibited by the Data Use Agreement allowing access to the data) and thus informed consent was not obtained.

## Results

A total of 10,040,850 deliveries, which after weighting corresponded to 18,712,085 deliveries nationwide on or after 24 weeks gestation, were identified (**Fig 1**). Among the weighted estimate of deliveries, 45,368 (0.5%) had HCV only; 84,582 (0.8%) had ICP only; and 1,967 had both HCV and ICP (<0.1%). Patients differed significantly in multiple factors (**Table 1**). Among the most prominent, substance use disorder was present in 69.2% of patients with HCV and 57.9% of patients with both HCV and ICP, but only in 4.6% of patients with ICP only and 6.3% of patients with neither disease. Similarly, tobacco use disorder was a comorbidity in 52.4% of patients with HCV and 42.8% of patients with both HCV and ICP, and only 3.8% in patients with ICP only and 5.3% of patients with neither disease. HIV was present in 0.7% of patients with HCV only and 0.8% of patients with both HCV and ICP, but only present in 0.1% of both patients with ICP only and neither disease.

In unadjusted analyses, the occurrence of most outcomes was notably higher for patients whose pregnancies were complicated by HCV or ICP when compared with those with neither complication (**Fig 2**). When pregnancies were complicated by either HCV or ICP, higher rates of SMM, cesarean delivery, preterm birth, and longer hospital lengths of stay were noted.

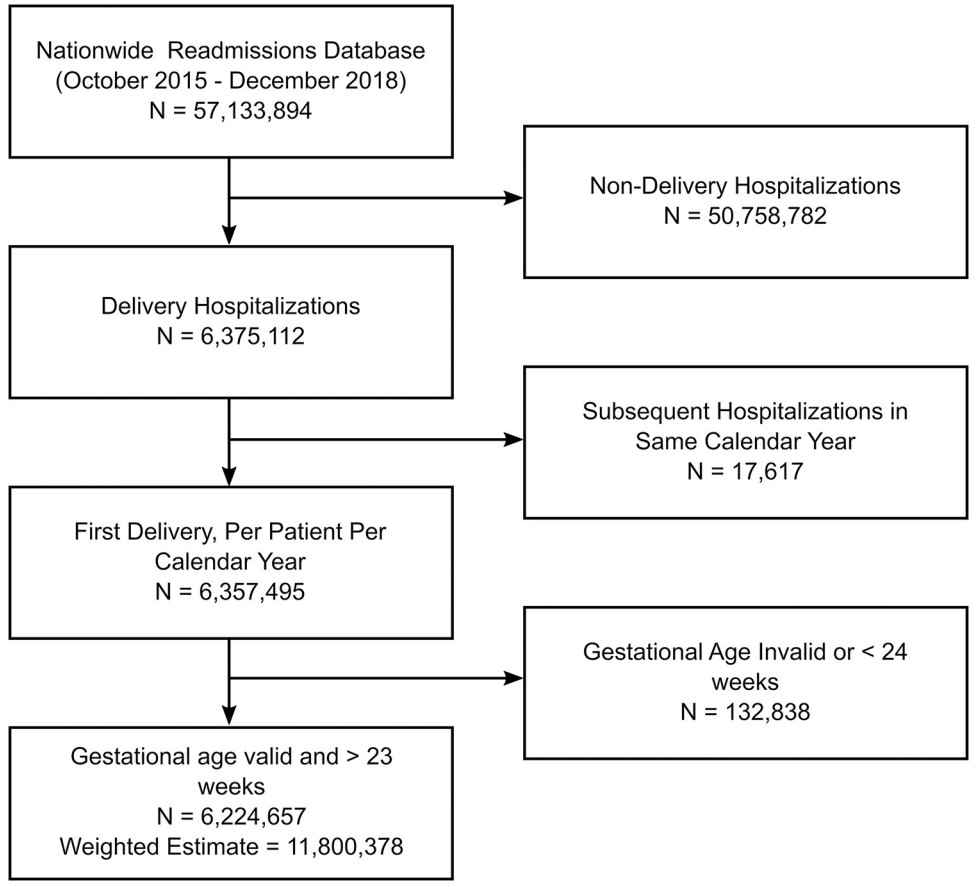

**Fig 1. CONSORT diagram delineating derivation of study cohort.**

**Table 1. Patient demographic and comorbidities.**

| | Hepatitis C / Cholestasis of Pregnancy Categories | | | | | |
| --- | --- | --- | --- | --- | --- | --- |
| | Overall (N = 10,040,850) (Weighted N = 18,712,085) | None (N = 9,908,933) (Weighted N = 18,471,505) | Hep C only (N = 45,368) (Weighted N = 88,269) | ICP Only (N = 84,582) (Weighted N = 148,569) | Hep C / ICP (N = 1,967) (Weighted N = 3,742) | p |
| | Mean (Standard Deviation) or % | | | | | |
| **Demographics** | | | | | | |
| Age in years at admission | 28.9 (5.8) | 28.9 (5.8) | 29.2 (4.9) | 29.3 (5.9) | 29.4 (5.3) | <0.001 |
| Zip Code Median Household Income | | | | | | <0.001 |
| Quartile 1 (Lowest) | 5,096,872 (27.4) | 5,028,336 (27.4) | 34,569 (39.6) | 32,731 (22.2) | 1,235 (33.4) | |
| Quartile 2 | 4,912,056 (26.4) | 4,845,629 (26.4) | 26,951 (30.9) | 38,384 (26.1) | 1,092 (29.6) | |
| Quartile 3 | 4,688,500 (25.2) | 4,629,469 (25.2) | 17,592 (20.2) | 40,610 (27.6) | 829 (22.4) | |
| Quartile 4 (Highest) | 3,886,823 (20.9) | 3,842,569 (20.9) | 8,137 (9.3) | 35,579 (24.2) | 538 (14.6) | |
| Primary expected payer (uniform) | | | | | | <0.001 |
| Medicare | 145,969 (0.8) | 143,009 (0.8) | 2,051 (2.3) | 828 (0.6) | 80 (2.1) | |
| Medicaid | 7,764,408 (41.5) | 7,626,982 (41.3) | 70,488 (80.0) | 63,965 (43.1) | 2,973 (79.4) | |
| Private | 9,944,669 (53.2) | 9,854,366 (53.4) | 12,228 (13.9) | 77,494 (52.2) | 582 (15.6) | |
| Self-pay | 280,129 (1.5) | 276,391 (1.5) | 1,554 (1.8) | 2,148 (1.4) | 36 (1.0) | |
| No charge | 9,109 (0.0) | 8,925 (0.0) | 59 (0.1) | 125 (0.1) | 0 (0.0) | |
| Other | 545,623 (2.9) | 539,869 (2.9) | 1,783 (2.0) | 3,899 (2.6) | 72 (1.9) | |
| **Comorbid Conditions** | | | | | | |
| Gestational Diabetes Mellitus | 1,473,679 (7.9) | 1,452,351 (7.9) | 4,565 (5.2) | 16,400 (11.0) | 362 (9.7) | <0.001 |
| HIV/AIDS | 17,290 (0.1) | 16,513 (0.1) | 617 (0.7) | 128 (0.1) | 32 (0.9) | <0.001 |
| Preexisting Diabetes Mellitus | 225,791 (1.2) | 222,564 (1.2) | 1,130 (1.3) | 2,048 (1.4) | 48 (1.3) | <0.001 |
| Prior cesarean birth | 3,294,368 (17.6) | 3,249,048 (17.6) | 19,727 (22.3) | 24,813 (16.7) | 780 (20.8) | <0.001 |
| Pulmonary hypertension | 4,699 (0.0) | 4,522 (0.0) | 115 (0.1) | 59 (0.0) | * | <0.001 |
| Multiple gestation | 332,437 (1.8) | 323,535 (1.8) | 1,634 (1.9) | 7,186 (4.8) | 82 (2.2) | <0.001 |
| Asthma | 1,035,956 (5.5) | 1,017,971 (5.5) | 8,225 (9.3) | 9,402 (6.3) | 358 (9.6) | <0.001 |
| Bleeding disorder | 421,002 (2.2) | 414,051 (2.2) | 2,372 (2.7) | 4,464 (3.0) | 114 (3.1) | <0.001 |
| BMI greater or equal to 40 | 514,730 (2.8) | 509,802 (2.8) | 1,718 (1.9) | 3,137 (2.1) | 74 (2.0) | <0.001 |
| Preexisting cardiac disease | 199,094 (1.1) | 195,003 (1.1) | 1,982 (2.2) | 2,011 (1.4) | 98 (2.6) | <0.001 |
| Chronic hypertension | 646,004 (3.5) | 638,100 (3.5) | 4,068 (4.6) | 3,677 (2.5) | 160 (4.3) | <0.001 |
| Chronic renal disease | 49,067 (0.3) | 47,864 (0.3) | 532 (0.6) | 654 (0.4) | 17 (0.5) | <0.001 |
| Connective tissue or autoimmune disease | 39,955 (0.2) | 39,060 (0.2) | 256 (0.3) | 621 (0.4) | 18 (0.5) | <0.001 |
| Placenta previa | 86,549 (0.5) | 85,446 (0.5) | 514 (0.6) | 573 (0.4) | 16 (0.4) | <0.001 |
| Preeclampsia with severe features | 657,679 (3.5) | 647,689 (3.5) | 3,776 (4.3) | 6,058 (4.1) | 157 (4.2) | <0.001 |
| Gestational hypertension / preeclampsia without severe features | 1,626,365 (8.7) | 1,606,938 (8.7) | 7,622 (8.6) | 11,452 (7.7) | 353 (9.4) | <0.001 |
| Substance use disorder | 1,230,773 (6.6) | 1,160,716 (6.3) | 61,102 (69.2) | 6,788 (4.6) | 2,167 (57.9) | <0.001 |
| Advanced maternal age | 3,321,211 (17.7) | 3,278,450 (17.7) | 13,516 (15.3) | 28,561 (19.2) | 684 (18.3) | <0.001 |
| Preexisting anemia | 2,540,594 (13.6) | 2,501,599 (13.5) | 13,749 (15.6) | 24,726 (16.6) | 520 (13.9) | <0.001 |
| Bariatric surgery | 55,6 the 35 (0.3) | 54,808 (0.3) | 182 (0.2) | 632 (0.4) | 13 (0.4) | <0.001 |
| Gastrointestinal disease | 1,080,084 (5.8) | 920,294 (5.0) | 7,479 (8.5) | 148,569 (100.0) | 3,742 (100.0) | <0.001 |
| Mental health disorder | 904,896 (4.8) | 876,438 (4.7) | 18,502 (21.0) | 9,090 (6.1) | 866 (23.1) | <0.001 |
| Neuromuscular disease | 95,453 (0.5) | 92,768 (0.5) | 1,741 (2.0) | 876 (0.6) | 68 (1.8) | <0.001 |
| Placental abruption | 198,809 (1.1) | 194,953 (1.1) | 2,508 (2.8) | 1,267 (0.9) | 81 (2.2) | <0.001 |
| Placenta accreta spectrum | 21,635 (0.1) | 21,274 (0.1) | 190 (0.2) | 165 (0.1) | * | <0.001 |
| Preterm birth | 1,832,350 (9.8) | 1,770,512 (9.6) | 16,149 (18.3) | 44,094 (29.7) | 1,596 (42.6) | <0.001 |

(*Continued*)

**Table 1.** (Continued)

| | Hepatitis C / Cholestasis of Pregnancy Categories | | | | | |
|---|---|---|---|---|---|---|
| | Overall (N = 10,040,850) (Weighted N = 18,712,085) | None (N = 9,908,933) (Weighted N = 18,471,505) | Hep C only (N = 45,368) (Weighted N = 88,269) | ICP Only (N = 84,582) (Weighted N = 148,569) | Hep C / ICP (N = 1,967) (Weighted N = 3,742) | p |
| | Mean (Standard Deviation) or % | | | | | |
| Tobacco Use Disorder | 1,040,331 (5.6) | 986,886 (5.3) | 46,238 (52.4) | 5,605 (3.8) | 1,602 (42.8) | <0.001 |
| Thyrotoxicosis | 51,378 (0.3) | 50,492 (0.3) | 303 (0.3) | 571 (0.4) | 12 (0.3) | <0.001 |
| **Hospital Characteristics** | | | | | | |
| Hospital Bedsize | | | | | | <0.001 |
| Small | 3,195,638 (17.1) | 3,157,363 (17.1) | 13,580 (15.4) | 24,154 (16.3) | 542 (14.5) | |
| Medium | 5,276,189 (28.2) | 5,211,676 (28.2) | 23,396 (26.5) | 40,163 (27.0) | 953 (25.5) | |
| Large | 10,240,258 (54.7) | 10,102,465 (54.7) | 51,292 (58.1) | 84,253 (56.7) | 2,247 (60.1) | |
| Hospital location / teaching status | | | | | | <0.001 |
| Metropolitan non-teaching | 3,808,898 (20.4) | 3,765,675 (20.4) | 15,763 (17.9) | 26,848 (18.1) | 612 (16.3) | |
| Metropolitan teaching | 13,164,075 (70.4) | 12,985,788 (70.3) | 61,879 (70.1) | 113,587 (76.5) | 2,822 (75.4) | |
| Non-metropolitan hospital | 1,739,112 (9.3) | 1,720,042 (9.3) | 10,627 (12.0) | 8,134 (5.5) | 309 (8.3) | |

P-values by weighted linear regression for continuous variables and weighted chi2 test for binary/categorical variables.

Missing values in Zip Code Income Quartile (44394 observations), and Primary Payor (6638 observations).

* Value 1–10 (inclusive) and suppressed by data vendor for patient privacy reasons.

HCV (but not ICP alone) was also associated with higher rates of ARDS and sepsis, while ICP (but not HCV alone) was also associated with higher rates of GDM. Patients with both HCV and ICP had higher rates of all outcomes, when compared with patients with neither condition.

When adjusted regression models were run including interaction terms between HCV and ICP, there was minimal evidence of antagonism or synergy (a greater or lesser effect from the combination of HCV and ICP than expected from the sum of the individual terms (**Fig 3**). The only statistically significant interaction terms were with GDM (OR 1.25, 95% CI 1.06, 1.48) and length of stay (OR 1.14, 95% CI 1.06–1.23), in which there appeared to be a small amount of synergy (patients stayed longer and had higher prevalence of GDM than expected from the contributions of HCV and ICP together). The regression models were repeated, removing all non-statistically significant interaction terms. (**Fig 4**). Given this adjustment, other than for length of stay and GDM, the odds of SMM and its components for patients with HCV and ICP were additive. In adjusted analyses, HCV and ICP were each associated with higher odds of SMM (OR 1.53, 95% CI 1.42–1.66), preterm birth (OR 5.09, 95% CI 4.87–5.33), and longer length of stay (rate ratio 1.46, 95% CI 1.35–1.58). HCV was also associated with higher odds of sepsis (OR 1.98, 95% CI 1.63–2.40), cesarean delivery (OR 1.13, 95% CI 1.10–1.16), and lower odds of GDM (OR 0.80, 95% CI 0.76–0.85). ICP was also associated with higher odds of acute kidney injury (OR 1.77, 95% CI 1.52–2.08) and GDM (OR 1.42, 95% CI 1.38–1.46).

## Discussion

This is the first study demonstrating the combined effects of HCV and ICP on maternal and obstetric outcomes. In our study, HCV correlated with an increased odds of SMM, sepsis, cesarean section, preterm birth, longer length of stay, as well as a decreased odds of GDM. ICP was associated with increased odds of SMM, acute kidney injury, GDM, preterm birth, and

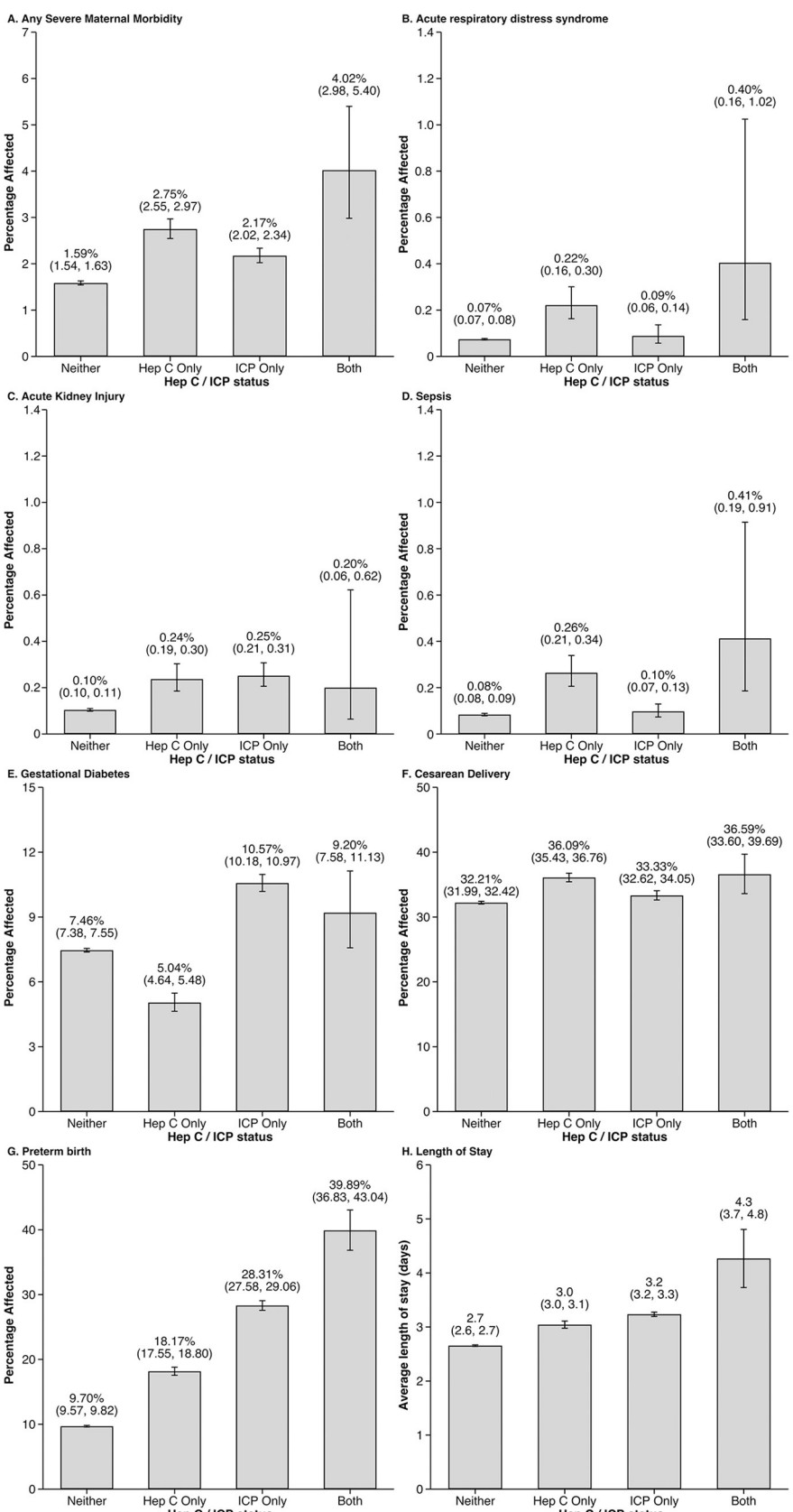

**Fig 2. Unadjusted rates of outcomes of interest in patients without Hepatitis C (HCV) nor Intrahepatic Cholestasis of Pregnancy (ICP), with HCV only, with ICP only, and with both.**

longer length of stay. There was no evidence of synergy in outcomes for HCV and ICP except for modest effects on GDM and length of stay.

## Impact of HCV on maternal and obstetric outcomes

There is mixed data on the association between HCV and preterm birth, Our findings that HCV is associated with an increased odds of preterm birth and cesarean section are consistent with prior studies that have shown an increased likelihood of preterm birth in this patient population [13,14], Another study demonstrated no association between HCV and preterm birth, in contrast to prior literature. Authors suggest that this discrepancy may be due to differences in definition of preterm birth used or use of chart abstraction for measurement of HCV as opposed to direct diagnostic testing [15]. Additionally, some prior studies have also shown that patients with HCV were more likely to undergo cesarean section [13]; however, cesarean section is not specifically recommended in patients with HCV as it has not been shown to decrease perinatal transmission [16].

With respect to maternal morbidity, extant literature suggests an association between HCV and hypertensive disorders of pregnancy [14,17], but previous studies have not evaluated overall SMM, Thus, our observation of the correlation between HCV and increased odds of overall SMM is novel. Interestingly, in our cohort HCV was associated with a decreased odds of GDM. We know that HCV interferes with the insulin-signaling pathway through multiple

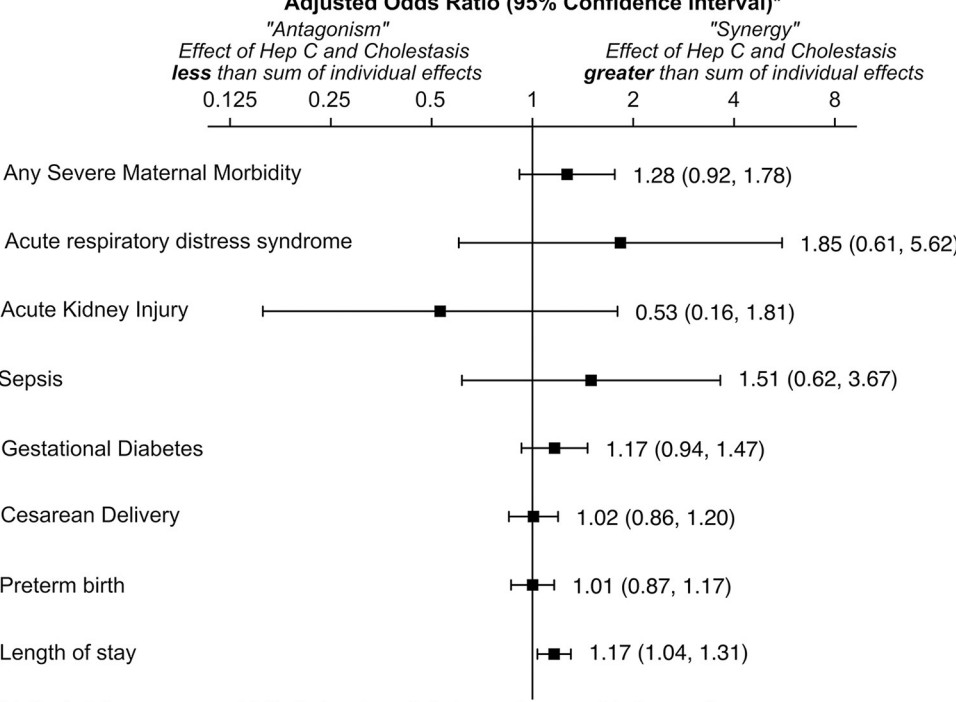

*Adjusted for age, comorbidity index, hospital size and ownership type, primary payor, ZIP code household income, and calendar year/quarter

**Fig 3. Adjusted regression analyses with interaction term for Hepatitis C (HCV) and Intrahepatic Cholestasis of Pregnancy (ICP), reported in odds ratios and 95% confidence intervals.**

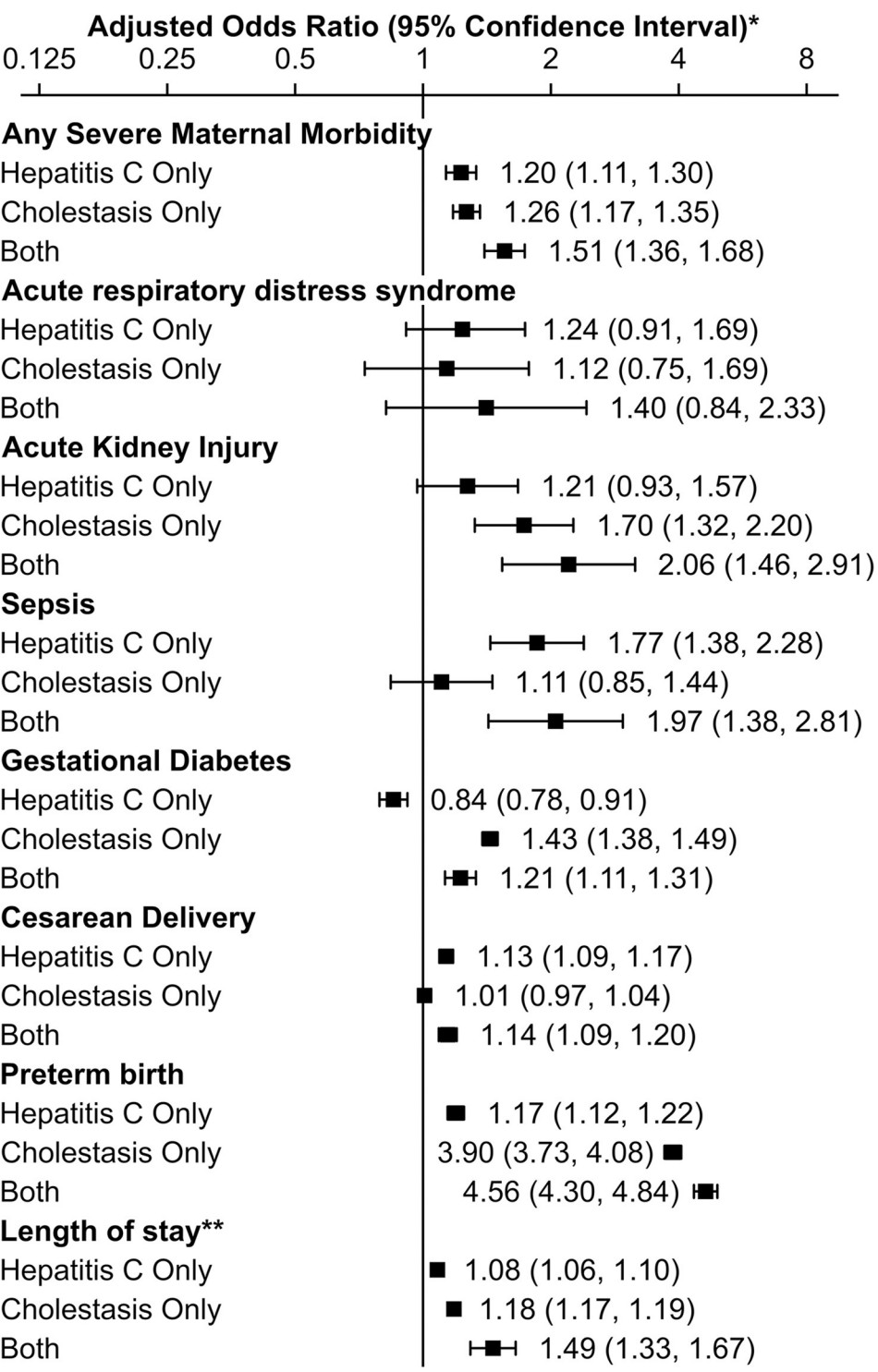

**Adjusted Odds Ratio (95% Confidence Interval)***

0.125   0.25   0.5   1   2   4   8

**Any Severe Maternal Morbidity**
Hepatitis C Only     1.20 (1.11, 1.30)
Cholestasis Only     1.26 (1.17, 1.35)
Both                 1.51 (1.36, 1.68)

**Acute respiratory distress syndrome**
Hepatitis C Only     1.24 (0.91, 1.69)
Cholestasis Only     1.12 (0.75, 1.69)
Both                 1.40 (0.84, 2.33)

**Acute Kidney Injury**
Hepatitis C Only     1.21 (0.93, 1.57)
Cholestasis Only     1.70 (1.32, 2.20)
Both                 2.06 (1.46, 2.91)

**Sepsis**
Hepatitis C Only     1.77 (1.38, 2.28)
Cholestasis Only     1.11 (0.85, 1.44)
Both                 1.97 (1.38, 2.81)

**Gestational Diabetes**
Hepatitis C Only     0.84 (0.78, 0.91)
Cholestasis Only     1.43 (1.38, 1.49)
Both                 1.21 (1.11, 1.31)

**Cesarean Delivery**
Hepatitis C Only     1.13 (1.09, 1.17)
Cholestasis Only     1.01 (0.97, 1.04)
Both                 1.14 (1.09, 1.20)

**Preterm birth**
Hepatitis C Only     1.17 (1.12, 1.22)
Cholestasis Only     3.90 (3.73, 4.08)
Both                 4.56 (4.30, 4.84)

**Length of stay****
Hepatitis C Only     1.08 (1.06, 1.10)
Cholestasis Only     1.18 (1.17, 1.19)
Both                 1.49 (1.33, 1.67)

* Compared to having neither Hepatitis C nor Cholestasis.
Adjusted for age, comorbidities, hospital size and ownership type,
primary payor, ZIP code  income, and calendar year/quarter
** Rate ratio rather than odds ratio, incorporating interaction term
between Hepatitis C and Cholestasis

**Fig 4. Effects of having Hepatitis C (HCV) only and Intrahepatic Cholestasis of Pregnancy (ICP) only, as well as the additive effect of having both, on the outcomes of interest, reported in odds ratios and 95% confidence intervals.**

mechanisms, can infect the beta islet cells, is associated with circulating micro-RNAs, and has been linked to type II diabetes mellitus [18]. Thus, it has been proposed that HCV infection could predispose patients to developing GDM [17,19]. Despite that proposed mechanism, findings in the literature are mixed. Reddick et al [20] queried the US Nationwide Inpatient Sample, and found increased adjusted odds for GDM among those with HCV. Notably, their data was from 1995–2005, so differences in coding or practice patterns may have contributed to the difference in outcomes. In contrast, a more recent inquiry of the Nationwide Inpatient Sample by Chen et al. using data from 2012–2018 found results in line with our study, observing lower odds of GDM among patients with HCV infection [14]. As our results add to an already heterogeneous picture, further epidemiologic and mechanistic study of HCV and GDM is needed.

## Impact of ICP on maternal outcomes

Although ICP has a well-established association with increased risk of poor fetal outcomes, it has been found to be a generally benign disease in terms of maternal outcomes. However, in some cases, especially in those with genetic alterations in bile composition and familial intrahepatic cholestasis of pregnancy, patients have a progressive course with chronic cholestasis, cholelithiasis portal hypertension, and liver failure [21]. It has also been associated with possible potentiation of the development of GDM through alterations in glucose homeostasis [22]. We found that there was also an increased odds of acute kidney injury, but there have been no studies, to our knowledge, that investigate or demonstrate a relationship between ICP and acute kidney injury. One case study described a patient who had ICP secondary to acute renal insufficiency in the setting of pyelonephritis; however, further research is needed to identify the mechanism behind this process [23]. While this may be attributed to under-ascertainment of cofounders that would lead to increased odds of kidney disease, current research has largely focused on fetal and neonatal outcomes as a result of ICP, and further research is needed to investigate the ICP effects on SMM, specifically whether there may be a relationship between ICP and kidney disease.

## Additive effects of ICP and HCV on maternal and obstetric outcomes

We found that having both HCV and ICP leads to an additive increased odds of SMM, preterm birth, and increased length of stay when compared to having either HCV, ICP, or neither. This is an important finding given that HCV is associated with increased odds of ICP. Many physiologic mechanisms have been proposed for how HCV leads to predisposition to ICP, with some research suggesting HCV impairs bile transportation, [24] potentially facilitating the occurrence of ICP at an earlier gestation age [8]. Moreover, certain mutations of biliary transporters, specifically the ABCB11 genotype, increase bile acids in infected patients [25] and this sub-group of HCV patients may be even more susceptible to ICP. Although the exact mechanism underlying the association between HCV and ICP has yet to be demonstrated, this study demonstrates that the mechanism is likely one that leads to additive effects rather than synergistic effects.

## Strengths and limitations

Our study is strengthened by the large, all-payor nature of the source dataset. Given that we are limited in what variables we can use by the database, we cannot know the serotype of HCV to stratify outcomes by serotype, or bile acid levels to characterize the severity of ICP. It was

also impossible to distinguish whether the patients received treatment for either disease that could have altered outcomes. However, the inability to control for these factors likely makes our results generalizable to patients with all types of HCV and severities of ICP. There are several limitations to administrative databases including reliability of coding data and lack of adequate control variables [26], but we are reassured on the former in that the rates of HCV and ICP are similar to the published literature in this cohort, as the reported prevalence of HCV in the Maternal Fetal Medicine Units Network is 0.3%. Given that HCV alone can cause cholestasis, it is possible some of the patients identified as having both HCV and ICP were misdiagnosed with ICP and rather had elevated bile acids and cholestasis attributable to HCV alone. This is a particular limitation in this dataset, as we cannot access clinical data beyond ICD-10 codes to assess how these diagnoses were made or if cholestasis resolved following pregnancy, as it would be expected were it caused by ICP.

While the models were adjusted for confounding factors using the Leonard comorbidity index, it is possible that HCV in this model is more of a "risk marker" rather than "risk predictor," as there are many social comorbidities that are strongly correlated with HCV, but undercoded in administrative data. It is also important that we clarify the nature of an additive effect in a logistic regression model, as was used here. In a logistic regression, covariates have a constant additive effect on the natural log of the odds of the event (or alternatively, a constant multiplicative effect on the odds of the event). We selected logistic regression given its known good performance properties for binary outcomes, whether common or rare, but it is possible that an alternative regression parameterization would produce different results. Finally, although ICP is known to increase risk of stillbirth, we were unable to assess this outcome well using this dataset. Because ICP management in the United States is designed to potentially reduce the rates of stillbirth through antenatal testing and planned preterm and early term deliveries, it would be challenging to ascertain whether differences present reflect underlying biological differences or differences in processes of care.

## Conclusions

This study demonstrates that having both HCV and ICP leads to an increased odds of SMM, preterm birth, and increased length of stay when compared to having either HCV, ICP, or both. These data may be useful in counseling patients regarding their increased risk of adverse outcomes when ICP presents in association with HCV versus ICP alone.

## Supporting information

**S1 Checklist.**
(DOCX)

**S2 Checklist. STROBE statement—Checklist of items that should be included in reports of observational studies.**
(DOCX)

**S1 Table. Reference list of ICD-10 codes used to identify patient characteristics.**
(DOCX)

## Acknowledgments

We greatly appreciate the assistance of Lamonica Daniels, CCS-P of the Duke University Health System Patient Revenue Management Organization in identifying the correct ICD-10-CM codes for the identification of intrahepatic cholestasis of pregnancy and Hepatitis C.

The authors appreciate the HCUP Data Partners who contribute data to the NRD. A complete list of partners can be found at (www.hcup-us.ahrq.gov/hcupdatapartners.jsp).

## Author Contributions

**Conceptualization:** Emily C. Goins, Lauren E. Wein, Virginia Y. Watkins, R. Phillips Heine, Brenna L. Hughes, Sarah K. Dotters-Katz, Jerome Jeffrey Federspiel.

**Data curation:** Lauren E. Wein, Sarah K. Dotters-Katz, Jerome Jeffrey Federspiel.

**Formal analysis:** Emily C. Goins, Lauren E. Wein, Virginia Y. Watkins, Alexa I. K. Campbell, R. Phillips Heine, Brenna L. Hughes, Sarah K. Dotters-Katz, Jerome Jeffrey Federspiel.

**Funding acquisition:** Brenna L. Hughes, Jerome Jeffrey Federspiel.

**Investigation:** Lauren E. Wein, Alexa I. K. Campbell, Sarah K. Dotters-Katz, Jerome Jeffrey Federspiel.

**Methodology:** Emily C. Goins, Lauren E. Wein, Virginia Y. Watkins, Alexa I. K. Campbell, Brenna L. Hughes, Sarah K. Dotters-Katz, Jerome Jeffrey Federspiel.

**Project administration:** Sarah K. Dotters-Katz, Jerome Jeffrey Federspiel.

**Resources:** Sarah K. Dotters-Katz, Jerome Jeffrey Federspiel.

**Software:** Jerome Jeffrey Federspiel.

**Visualization:** Jerome Jeffrey Federspiel.

**Writing – original draft:** Emily C. Goins, Lauren E. Wein, Brenna L. Hughes, Sarah K. Dotters-Katz, Jerome Jeffrey Federspiel.

**Writing – review & editing:** Lauren E. Wein, Virginia Y. Watkins, Alexa I. K. Campbell, R. Phillips Heine, Brenna L. Hughes, Sarah K. Dotters-Katz, Jerome Jeffrey Federspiel.

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
