## [Decision Letter · Decision Letter 0]

26 Jun 2023

PONE-D-23-06302Maternal and neonatal outcomes in patients with hepatitis C and intrahepatic cholestasis of pregnancy: the sum of the partsPLOS ONE

Dear Dr. Jeromy,

Thank you for submitting your manuscript to PLOS ONE. After careful consideration, we feel that it has merit but does not fully meet PLOS ONE’s publication criteria as it currently stands. Therefore, we invite you to submit a revised version of the manuscript that addresses the points raised during the review process.

We look forward to receiving your revised manuscript.

Kind regards,

Surangi Jayakody, MBBS, MSc, MD

Academic Editor

PLOS ONE

Journal Requirements:

“).  Work contained in this manuscript were made possible by the following grants from the National Institutes of Health (K24-AI093969 [VGF]; NIAID (T32-AI100851 [EME]); and TL1-TR002555 [JJF]). Data acquisition was also supported by funding from the Foundation for Women and Girls with Blood Disorders to JJF.  “

“The work contained in this manuscript were made possible by the following grants from the National Institutes of Health (TL1-TR002555 [JJF] and K12-HD103083). Data acquisition was also supported by funding from the Foundation for Women and Girls with Blood Disorders to JJF. The funders had no role in study design, data collection and analysis, decision to publish, or preparation of the manuscript.”

Reviewers' comments:

Reviewer's Responses to Questions

**Comments to the Author**

1. Is the manuscript technically sound, and do the data support the conclusions?

Reviewer #1: Yes

Reviewer #2: Yes

Reviewer #3: Yes

2. Has the statistical analysis been performed appropriately and rigorously? 

Reviewer #1: Yes

Reviewer #2: Yes

Reviewer #3: Yes

3. Have the authors made all data underlying the findings in their manuscript fully available?

Reviewer #1: Yes

Reviewer #2: Yes

Reviewer #3: No

4. Is the manuscript presented in an intelligible fashion and written in standard English?

Reviewer #1: Yes

Reviewer #2: Yes

Reviewer #3: Yes

5. Review Comments to the Author

Reviewer #1: Your project is a reflection of your hard work and dedication, and I appreciate both.

Once you updated the majority of the outdated references, I'll gladly reconsider your work. Five references older than five years are permitted, as I have noticed twenty of your cited works are more than five years out of date.

Reviewer #2: This is a good article which gives clear picture about out come of Hep C and ICP in pregnancy. But it is important to know whether ICP is in disease process of Hep C. Also it is important to analysis treatment modalities in this patients. When we consider the morbidity its important to find number of patients need specialized care such as ICU facility.

When consider the out come I think better to investigate about placental accidents if possible in your cohort. Because its common with ICP in pregnancy.

Reviewer #3: I wish to thank the authors for the study of a very interesting observation. I have few comments,

HCV is considered a risk factor for ICP.

"Several studies have linked HCV infection to a higher risk of developing ICP. However some data demonstrates contradictory results."

"HCV-associated cholestasis is also well described in different reports on liver transplantation. Its occurrence is attributed to viral overload and suppression of host immune response. The mechanisms of HCV-associated pruritus are attributed to HCV-induced cholestasis and the induction of interferon-stimulated genes (ISGs) as a result of viral overload."

ICP by definition is pregnancy limited and resolves after delivery. HCV causing cholestasis may not.

It would be interesting if the authors could further describe their population, considering the disease resolution in ICP+HCV group (and the HCV group) to describe the bias (if any) by including HCV cases with none resolution of cholestasis following pregnancy (if any).

If the data is not available, suggest authors discuss this limitation with the other study limitations.

6. PLOS authors have the option to publish the peer review history of their article (what does this mean?). If published, this will include your full peer review and any attached files.

Reviewer #1: **Yes: **Mena Abdalla

Reviewer #2: No

Reviewer #3: **Yes: **Indu Asanka Jayawardane

---

## [Author Response · Author response to Decision Letter 0]

20 Jul 2023

Journal/Article Identifying No.

Article Title

Response to Reviewers

We use a response format common in our field to respond to reviewer comments:

a. Comment from reviewer

b. Our response

c. Line numbers of changes

d. Change made in manuscript

Reviewer 1, Comment 1

a. Once you updated the majority of the outdated references, I'll gladly reconsider your work. Five references older than five years are permitted, as I have noticed twenty of your cited works are more than five years out of date.

b. Thank you for highlighting this. We have reviewed our references and updated our out-of-date references as much as possible. Unfortunately, this topic is one which has, despite its importance, received less contemporary attention than it deserves. After an exhaustive search, we were able to reduce the number of older citations from 19 to 9. 

c. Throughout manuscript/reference list

d. See tracked changes

Reviewer 2, Comment 1

a. This is a good article which gives clear picture about outcome of Hep C and ICP in pregnancy. But it is important to know whether ICP is in disease process of Hep C. 

b. Thank you for this comment. We agree that it is important to better understand the relationship between ICP and Hep C to determine what is driving their association. Although we are limited in our ability to assess this relationship given the cross-sectional nature of this study, we have added language to the introduction to re-emphasize this point, in addition to the language already present in the discussion.

c. Line numbers 94-97

d. “The mechanism behind this association is unclear, with some research suggesting HCV may directly increase bile acids in birthing people, and some suggesting a more indirect role through alteration of bile acid metabolism.”

Reviewer 2, Comment 2

a. Also it is important to analysis treatment modalities in this patients. 

b. This is a great point and one of the unfortunate limitations of the dataset from which these data are drawn. Although we would like to include treatment received in our analysis, the Nationwide Readmissions Database limited dataset is limited to hospital level data, demographics, and ICD-10 codes. This point is mentioned in lines 275-278 of the manuscript.

c. No changes

d. N/A

Reviewer 2, Comment 3

a. When we consider the morbidity it’s important to find number of patients need specialized care such as ICU facility.

b. We agree that it is important to assess degree of morbidity and need for specialized care in patients through variables such as ICU admissions. While we did not include this variable specifically, many of the variables included still address this point. We used severe maternal morbidity (SMM) given that many of the included indicators of SMM require a high level of care. Moreover, we included sepsis, ARDS, and length of stay as secondary outcomes to assess if outcomes in these more specific indicators mirror those observed in SMM. We feel that our included variables adequately characterize how HCV and ICP alone and in tandem impact maternal outcomes and illustrate the severity of these outcomes.

c. No changes

d. N/A

Reviewer 2, Comment 4

a. When consider the outcome I think better to investigate about placental accidents if possible in your cohort. Because its common with ICP in pregnancy.

b. Thank you for this comment. We agree that the pathogenesis and morbidity of ICP in relation to the placenta is an important area of future study, particularly given concern for placental response to bile acids as a cause of stillbirth. While we do not have data assess placental function in detail, we were able to assess incidence of certain placental conditions, including placenta accrete spectrum, abruption, and previa, as shown in Table 1. These data demonstrate a lower incidence of abruption in ICP and higher in Hep C; however, this incidence may be artificially lowered due to iatrogenic early delivery for ICP, given this is the standard of care for many patients with ICP in the United States. We decided not to focus on outcomes such as placental abruption given this potential for bias.

c. No changes

d. N/A

Reviewer 3, Comment 1

a. HCV is considered a risk factor for ICP. "Several studies have linked HCV infection to a higher risk of developing ICP. However some data demonstrates contradictory results."

"HCV-associated cholestasis is also well described in different reports on liver transplantation. Its occurrence is attributed to viral overload and suppression of host immune response. The mechanisms of HCV-associated pruritus are attributed to HCV-induced cholestasis and the induction of interferon-stimulated genes (ISGs) as a result of viral overload." ICP by definition is pregnancy limited and resolves after delivery. HCV causing cholestasis may not. It would be interesting if the authors could further describe their population, considering the disease resolution in ICP+HCV group (and the HCV group) to describe the bias (if any) by including HCV cases with none resolution of cholestasis following pregnancy (if any).

If the data is not available, suggest authors discuss this limitation with the other study limitations.

b. Thank you for this suggestion. The Nationwide Readmissions Database is limited in ability to adequately follow-up patients as described in the methods, so we could not identify patients in which cholestasis resolved following pregnancy. However, we agree this is a limitation to this study, and have amended our limitations section to reflect this.

c. Lines 265-268

d. “Given that HCV alone can cause cholestasis, it is possible some of the patients identified as having both HCV and ICP were misdiagnosed with ICP and rather had elevated bile acids and cholestasis attributable to HCV alone. This is a particular limitation in this dataset, as we cannot access clinical data beyond ICD-10 codes to assess how these diagnoses were made or if cholestasis resolved following pregnancy, as it would be expected to were it caused by ICP.”

---

## [Decision Letter · Decision Letter 1]

4 Oct 2023

Maternal and neonatal outcomes in patients with hepatitis C and intrahepatic cholestasis of pregnancy: the sum of the parts

PONE-D-23-06302R1

Dear Dr. Federspiel,

We’re pleased to inform you that your manuscript has been judged scientifically suitable for publication and will be formally accepted for publication once it meets all outstanding technical requirements.

Kind regards,

Victor Daniel Miron

Academic Editor

PLOS ONE

Additional Editor Comments (optional):

Reviewers' comments:

Reviewer's Responses to Questions

**Comments to the Author**

1. If the authors have adequately addressed your comments raised in a previous round of review and you feel that this manuscript is now acceptable for publication, you may indicate that here to bypass the “Comments to the Author” section, enter your conflict of interest statement in the “Confidential to Editor” section, and submit your "Accept" recommendation.

Reviewer #1: All comments have been addressed

Reviewer #3: All comments have been addressed

2. Is the manuscript technically sound, and do the data support the conclusions?

Reviewer #1: Yes

Reviewer #3: Yes

3. Has the statistical analysis been performed appropriately and rigorously? 

Reviewer #1: Yes

Reviewer #3: Yes

4. Have the authors made all data underlying the findings in their manuscript fully available?

Reviewer #1: Yes

Reviewer #3: Yes

5. Is the manuscript presented in an intelligible fashion and written in standard English?

Reviewer #1: Yes

Reviewer #3: Yes

6. Review Comments to the Author

Reviewer #1: Thank you for the authors for addressing the previous reviewers' comments.

From my side, I find it fair to accept your interesting article, all the best.

Reviewer #3: Congratulations. Data source limitations were added by the authors in limitations. Authors have addressed the review concerns. Recommendation- Accept.

7. PLOS authors have the option to publish the peer review history of their article (what does this mean?). If published, this will include your full peer review and any attached files.

Reviewer #1: **Yes: **Mena Abdalla

Reviewer #3: **Yes: **Indu Asanka Jayawardane

---

## [Editor Report · Acceptance letter]

10 Oct 2023

PONE-D-23-06302R1 

Maternal and neonatal outcomes in patients with hepatitis C and intrahepatic cholestasis of pregnancy: the sum of the parts 

Dear Dr. Federspiel:

I'm pleased to inform you that your manuscript has been deemed suitable for publication in PLOS ONE. Congratulations! Your manuscript is now with our production department. 

Kind regards, 

on behalf of

Dr. Victor Daniel Miron 

Academic Editor

PLOS ONE